# Early mucosal events promote distinct mucosal and systemic antibody responses to live attenuated influenza vaccine

Ryan S. Thwaites [1] ✉, Ashley S. S. Uruchurtu [1], Victor Augusti Negri[2], Megan E. Cole [3], Nehmat Singh[1], Nelisa Poshai[1], David Jackson[4], Katja Hoschler[4], Tina Baker[2], Ian C. Scott[2], Xavier Romero Ros[5], Emma Suzanne Cohen[5], Maria Zambon [4], Katrina M. Pollock [3], Trevor T. Hansel[1] & Peter J. M. Openshaw [1] ✉

Compared to intramuscular vaccines, nasally administered vaccines have the advantage of inducing local mucosal immune responses that may block infection and interrupt transmission of respiratory pathogens. Live attenuated influenza vaccine (LAIV) is effective in preventing influenza in children, but a correlate of protection for LAIV remains unclear. Studying young adult volunteers, we observe that LAIV induces distinct, compartmentalized, antibody responses in the mucosa and blood. Seeking immunologic correlates of these distinct antibody responses we find associations with mucosal IL-33 release in the first 8 hours post-inoculation and divergent CD8$^+$ and circulating T follicular helper (cTfh) T cell responses 7 days post-inoculation. Mucosal antibodies are induced separately from blood antibodies, are associated with distinct immune responses early post-inoculation, and may provide a correlate of protection for mucosal vaccination. This study was registered as NCT04110366 and reports primary (mucosal antibody) and secondary (blood antibody, and nasal viral load and cytokine) endpoint data.

Influenza infections are estimated to result in up to 500,000 deaths per annum in seasonal epidemics, and pose a constant pandemic threat[1]. Two influenza types cause respiratory infections in humans, termed influenza A and influenza B, the constant antigenic evolution of which serves to evade host immunity[1]. Annual reformulation of influenza vaccines aims to match antigenic drift, and these vaccines are generally highly efficacious when well matched[1,2]. Antigenic evolution particularly effects the viral surface neuraminidase and haemagglutinin proteins that are targeted by host immunity[1,2]. The most widely used influenza vaccines are intramuscularly delivered inactivated virus, which trigger the generation of high titre neutralizing antibodies that are one correlate of protection against influenza like illness (ILI)[2,3].

These antibodies are commonly measured in blood using haemagglutinin inhibition assays (HAI), though cellular immunity and antibodies against a wider array of antigenic targets and antibody effector properties are also thought to contribute to protection[3].

Nasally delivered live attenuated influenza vaccine (LAIV) is more efficacious than intramuscular vaccination for protecting children from influenza[4] and is prioritized in children in some countries, including the UK[5]. The effectiveness of LAIV declines with age[6], which may be attributed to the gradual accumulation of homosubtypic or heterosubtypic immunity. The replication of each of the 4 constituent vaccine viruses in LAIV has been considered necessary to induce protective responses[6,7]. However, the requirement for detectable vaccine

[1]National Heart and Lung Institute, Imperial College London, London, UK. [2]Translational Science and Experimental Medicine, Research and Early Development, Respiratory and Immunology, BioPharmaceuticals R&D, AstraZeneca, Cambridge, UK. [3]Department of Infectious Disease, Imperial College London, London, UK. [4]United Kingdom Health Security Agency, London, UK. [5]Bioscience Asthma and Skin Immunity, Research and Early Development, Respiratory and Immunology, BioPharmaceuticals R&D, AstraZeneca, Cambridge, UK. ✉e-mail: r.thwaites@imperial.ac.uk; p.openshaw@imperial.ac.uk

replication to induce immune responses has been questioned and remains unclear[8]. Compared to intramuscular vaccines, nasally administered vaccines have the additional advantage of inducing local mucosal immune responses that may block infection and interrupt transmission of respiratory pathogens[9]. This effect was apparent in trials of LAIV, where adult ILI cases decreased following vaccination of day care and school age children with LAIV[10,11]. These indirect effects of mucosal vaccines may be crucial in limiting pathogen transmission and there is an unmet need for mucosal vaccines against SARS-CoV-2[12].

Immune responses in the respiratory mucosa are compartmentalized from those of the systemic immune system and other mucosal tissues, with local responses dependent on mucosa-associated lymphoid tissues[13,14]. The immunological processes underpinning compartmentalization of local and systemic responses are thought to depend on antigen drainage to mucosal or non-mucosal lymphoid tissues and the environmentally influenced phenotypic and functional distinctions of mucosal lymphocytes[13–15].

Despite its demonstrable efficacy against influenza in children, a correlate of protection for LAIV remains elusive[16]. Blood HAI responses to LAIV are weaker than intramuscular vaccines, but LAIV is more likely to trigger increases in mucosal IgA titres[17,18]. There is evidence that mucosal and systemic antibody responses are separately regulated, and that local mucosal production of IgA could result from immune process distinct from those generating peripheral blood antibodies[15,17,19]. LAIV vaccination has been documented to result in blood antibody responses against H1N1 in just 9% of participants when measured by HAI or 24% of participants when measured by immunohistochemistry, while nasal IgA responses were seen in 33% of participants[18]. While mucosal IgA is considered an important result of LAIV vaccination[18], the mechanisms underpinning mucosal antibody responses to LAIV are unclear.

In this work, we sought to understand the nature of the systemic and mucosal humoral response to LAIV, and the mechanisms behind these responses. We inoculated healthy young adults ($n = 40$) with LAIV and performed detailed analyses of immune activation after vaccination. Our data indicated that mucosal immune responses in the first hours post-inoculation were associated with the scale of LAIV replication in the airway. Vaccination induced interferon (IFN) dominated mucosal immune responses, irrespective of detectable viral replication, in addition to distinct systemic and mucosal antibody responses. This compartmentalization of antibody responses was associated with different profiles of systemic and mucosal immune activation and viral replication.

## Results

### LAIV inoculation and replication

We recruited 40 healthy adults (median age 22 years (range 19–29), 9 males; Supplementary Table 1) and collected baseline blood, nasosorption, and nasal curettage samples 7 days prior to inoculation with quadrivalent LAIV (AstraZeneca, 2018/19 formulation) between October 2018 and April 2019. On the day of inoculation (study day (SD) 0) nasosorption samples were collected hourly between 3–8 h post inoculation (p.i.). Further nasal and blood samples were collected from participants 24 h (SD1), 72 h (SD3) and 168 h (SD7) p.i. and on SD28 (Fig. 1a). No nasal obstruction or other symptoms were reported (Supplementary Fig. 1).

Replication of each of the 4 LAIV constituent vaccine viruses (A/Michigan/45/2015 (H1N1)pdm09-like virus ("H1"), A/Singapore/INFIMH-16-0019/2016 A(H3N2)-like virus ("H3"), a B/Phuket/3073/2013-like B/Yamagata-lineage virus ("B/Yam") and a B/Colorado/06/2017-like B/Victoria-lineage virus ("B/Vic")) was quantified from nasosorption samples collected on SD1, 3, and 7 using RT-qPCR. H1 and H3 were only detected in one sample each (both at SD1) while >50% of participants were positive for B/Yam and/or B/Vic for at least one timepoint (24/40 Fig. 1b; 23/40 Fig. 1c, respectively). This lack of influenza A vaccine virus shedding was reminiscent of our observations in vaccinated children where shedding of H1 and H3 components was less frequent and at lower scale than influenza B viruses[20]. The scale and duration of vaccine shedding was variable between participants, with little concordance in positivity for shedding between influenza B viruses, shown as area under curve (AUC) viral shedding per day (Fig. 1d).

### Mucosal immune responses to LAIV

We next sought to profile the mucosal immune response to LAIV by quantifying cytokine and chemokine immune mediators from nasosorption eluates. Levels of IFNλ were above baseline 72 h p.i. and remained elevated at SD7 (168 h p.i.; both $P < 0.05$, Fig. 2a). IFNγ was elevated at 24 h and 72 h p.i. (both $P < 0.05$, Fig. 2b), while the IFN-induced chemokine CXCL10 and IL-6 were elevated 72 h p.i. (both $P < 0.05$, Fig. 2c, 2d, respectively). In contrast to these inductions at 24–72 h p.i., the epithelial alarmin IL-33 was elevated above baseline within 8 h p.i. but returned to baseline levels by 24 h p.i. (Fig. 2e). To

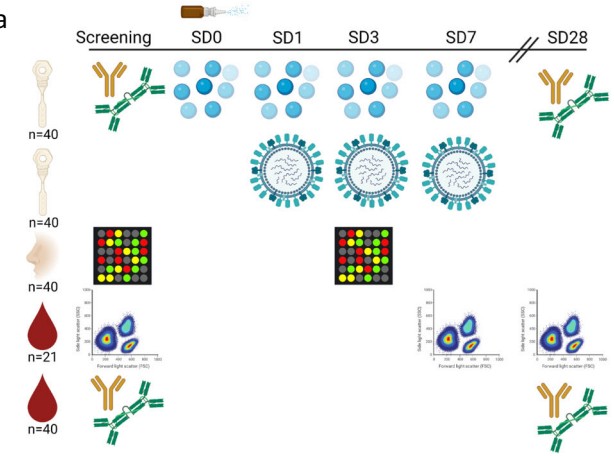

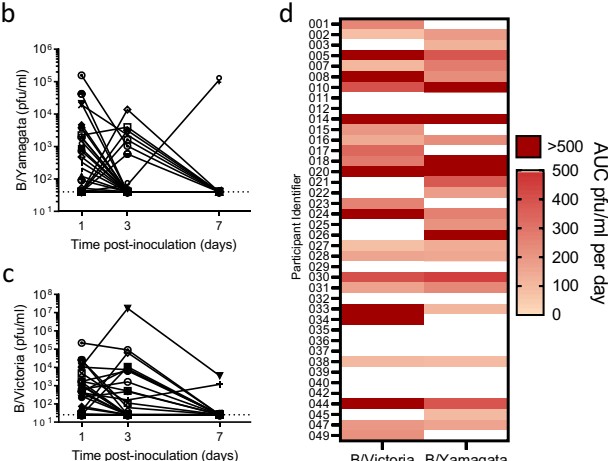

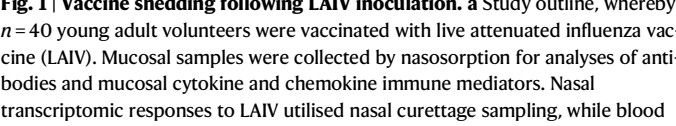

**Fig. 1 | Vaccine shedding following LAIV inoculation. a** Study outline, whereby $n = 40$ young adult volunteers were vaccinated with live attenuated influenza vaccine (LAIV). Mucosal samples were collected by nasosorption for analyses of antibodies and mucosal cytokine and chemokine immune mediators. Nasal transcriptomic responses to LAIV utilised nasal curettage sampling, while blood samples were utilised for peripheral cytometry and antibody analyses. Shedding of vaccine viruses was quantified from nasosorption samples for **b** B/Yamagata and **c** B/Victoria. **d** Heatmap of area under curve (AUC) of vaccine virus shedding in all 40 volunteers. Pfu/ml=plaque forming units per millilitre. Panel **a** was created with Biorender.com. Source data are provided as a Source Data file.

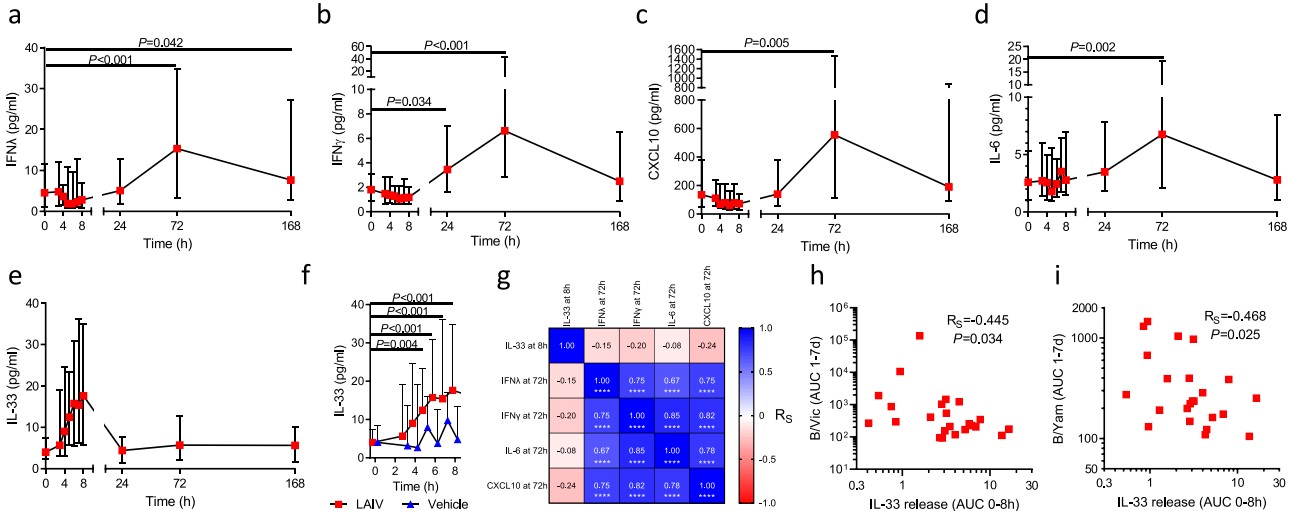

**Fig. 2 | Mucosal cytokine and chemokine responses following LAIV inoculation.** Immune mediators in the nasal site of vaccination were quantified from naso-sorption samples collected over a time course post-vaccination for **a** IFNλ/IL-29, **b** IFNγ, **c** CXCL10, **d** IL-6, and **e** IL-33. **f** IL-33 levels over the first 8 hours post-LAIV (red, n = 40) and post vehicle control (blue, n = 8). **g** Correlation matrix of nasal mediator responses to LAIV at the peak time point for each mediator. Correlations between nasal IL-33 responses, measured as area-under-curve (AUC) of 0-8 h post-LAIV, versus later (1-7 days post-LAIV) AUC of vaccine shedding for **h** B/Victoria, and **i** B/Yamagata. Panels **a–f** show median and interquartile range values. In **a–d** and **f**, significance was tested between each time point and baseline (0 h) using two-sided Friedman tests with Dunn's multiple comparison tests. Correlations in **g–i** used two-sided Spearman R-values and P-value adjustment for multiple testing where the significance of correlations is denoted by asterisks in **g**. \*P < 0.05, \*\*P < 0.01, \*\*\*P < 0.001 \*\*\*\*P < 0.0001. Source data are provided as a Source Data file.

confirm that IL-33 release was not a result of repeated sampling of the nasal mucosa a vehicle control challenge was conducted on 8 partici-pants. This confirmed stable IL-33 levels in the absence of LAIV, while IL-33 was significantly elevated above baseline in the LAIV group from 5 h p.i. (Fig. 2f). Correlation matrix analysis determined that levels of IFNλ, IFNγ, IL-6, and CXCL10 at 72 h p.i. (the peak of these responses) were closely correlated but were not associated with the scale of IL-33 release at 8 h p.i. (Fig. 2g). We sought to determine whether mediator release p.i. was associated with the scale of subsequent vaccine shed-ding. Significant inverse correlations were evident between early (3–8 h p.i.) IL-33 induction and shedding of both B/Vic and B/Yam (Fig. 2h and i, respectively) while AUC IFNλ, IFNγ, IL-6, and CXCL10 levels were not associated with virus shedding. Despite the docu-mented role of IL-33 in triggering type-2 immune responses[21], no induction of IL-13 was observed at any time point. Thus, an IFN-related response to LAIV is evident in the nasal mucosa following LAIV inoculation, preceded by an earlier release of IL-33 which is not asso-ciated with the subsequent IFN-response but is inversely related to viral replication.

Given the apparent suppression of viral replication occurred independently of the IFN response and was inversely related to early IL-33 release following LAIV inoculation, we sought to identify molecular pathways associated with viral replication. Nasal curettage samples from baseline and SD3 of all participants were analyzed by RNA-Seq, identifying 130 upregulated differentially expressed genes (DEGs) between SD3 and baseline including the IFN-induced chemokines *CXCL9/10/11* (Supplementary Fig. 2A). Biological processes (BP) enrichment analysis identified several upregulated pathways typically associated with anti-viral immune responses, including "Cytokine-mediated signaling pathway" and "Defense response to virus" (Sup-plementary Fig. 2B).

We next sought to study differences in DEGs between participants that shed virus at any timepoint ("Shedders", n = 20) and those that did not ("Non-Shedders", n = 5). At SD0, 118 DEGs were evident between Shedders and Non-Shedders (Supplementary Fig. 2C), though enrich-ment analysis did not identify any significantly altered pathways amongst these DEGs. At SD3 only 5 DEGs were evident between Shedders and Non-Shedders (Supplementary Fig. 2D). We reasoned

from these data that the small number of Non-Shedders (n = 5) in this analysis may prevent meaningful direct comparisons between groups at each timepoint. Instead, we compared individuals within the Shed-ders or Non-Shedders group to their individual SD0 baseline value, to better account for heterogeneity between individuals. The response amongst Shedders included 21 DEGs, including *CXCL10/11* (Fig. 3a), while BP enrichment analysis indicated the response was dominated by immune signaling and T cell chemotaxis (Fig. 3b). By contrast, the DEGs amongst Non-Shedders between SD3 and baseline were largely distinct from Shedders (Fig. 3c), with fewer terms relating to T cell responses but similar evidence of neutrophilic responses (Fig. 3d). To further study these differences in DEGs between Shedders and Non-Shedders we looked at some key DEGs, namely *CXCL9, CXCL10, SOCS1*, and *CXCR1* (Fig. 3e–h, respectively). This indicated that transcriptional responses to inoculation were similar in Shedders and Non-Shedders, though *CXCL10* induction was marginally stronger in Shedders. Shedders also had greater baseline (SD0) expression of the neutrophil associated chemokine receptor *CXCR1*, in agreement with our previous observation of neutrophil mediated enhancement of viral infection[22].

In agreement with the transcriptional data, median protein CXCL10 levels were higher in Shedders, relative to Non-Shedders, at SD3 (Fig. 3i), though this difference was not significant. Levels of the neutrophil chemoattractant CXCL8 (a ligand for CXCR1) and IL-33 were equivalent between groups at all timepoints (Fig. 3j, 3k, respectively). Together, these data indicated that a prototypical antiviral immune response, including IFNγ, IFNλ (IL-29) and numerous ISGs, is evident in the nasal mucosa within days of LAIV inoculation. This IFN-led response was evident irrespective of observable viral shedding, but was most pro-nounced when vaccine replication is evident.

## Distinct mucosal and systemic antibody responses to LAIV
LAIV replication and shedding has been considered a pre-requisite for humoral immune responses[7,23]. Despite the absence of discernable shedding for H1 and H3, there were elevated titres of antibodies against all 4 vaccine component viruses at SD28, in both blood (IgG Fig. 4a; IgA Supplementary Fig. 3) and nasosorption eluates (IgA Fig. 4b; IgG Supplementary Fig. 3). Using a 4-fold rise cut-off, 18/40 (45%) of participants had serum IgG responses to at least 1 LAIV

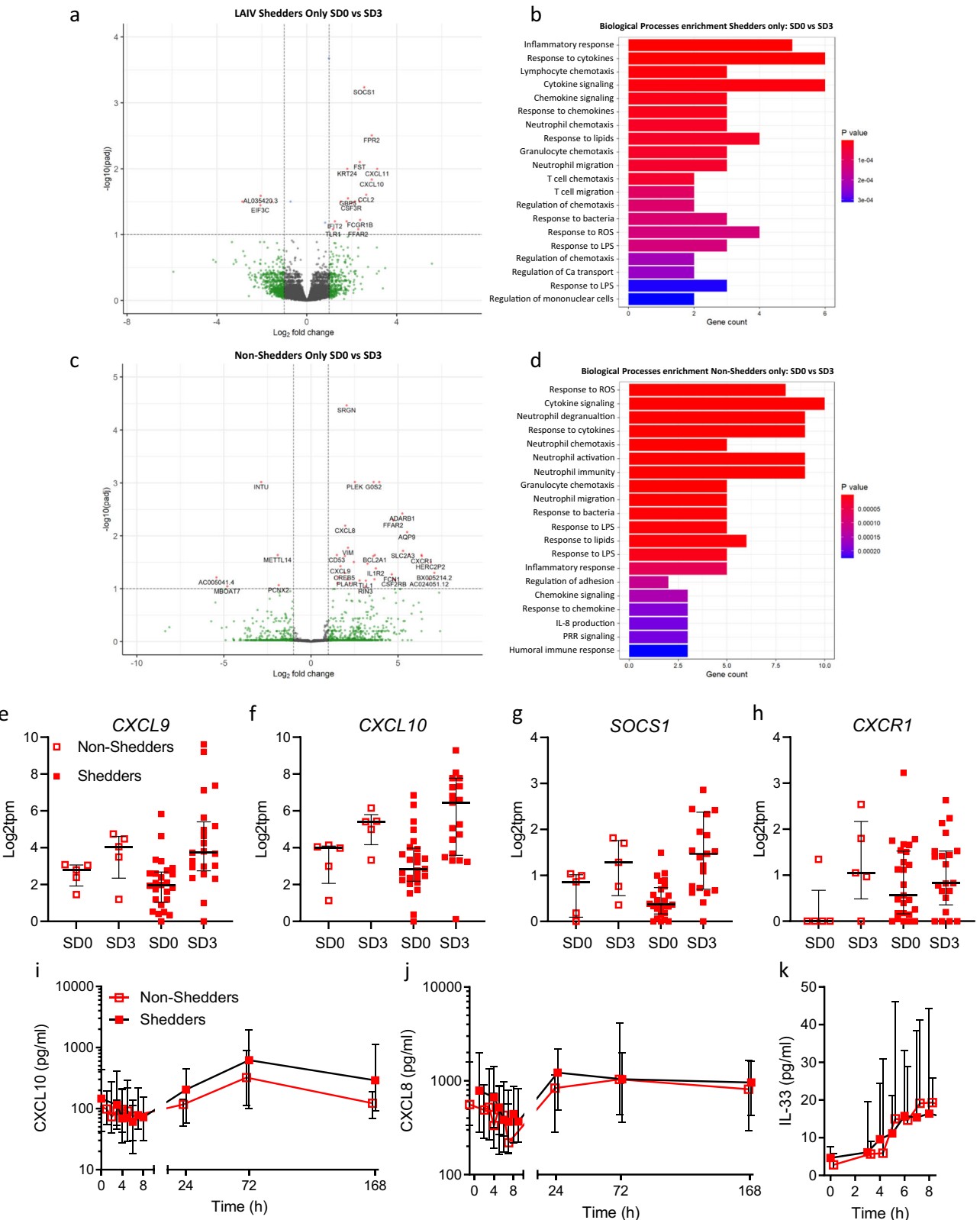

component while 21/40 (53%) participants had nasal IgA responses (Fig. 4c). Increases in blood neutralizing antibody, as measured by hemagglutinin-inhibition assays (HAI), were evident amongst participants with low HAI titres at SD0 (Fig. 4d, Supplementary Fig. 4). Nasal IgA titres increased between SD0 and SD28 irrespective of LAIV shedding (Supplementary Fig. 5A); however, significant induction of serum IgG was only evident in Shedders and not Non-Shedders

(Supplementary Fig. 5B). Shedders also had lower levels of serum IgG anti-B/Vic ($P = 0.02$, Supplementary Fig. 5B), but nasal IgA titres were equivalent between Shedders and Non-Shedders at SD0 (Supplementary Fig. 5A). These results re-enforce the documented ability of LAIV to induce seroconversion in adults with low pre-existing humoral immunity[24] and indicate that nasal antibody responses show less association with vaccine shedding.

**Fig. 3 | Robust mucosal transcriptomic anti-viral response to vaccination irrespective of vaccine shedding.** RNA-Sequencing was conducted on nasal tissue samples collected at baseline (SD0, $n = 34$) and at 72 h post-vaccination (SD3, $n = 26$). **a** Volcano plot of differentially expressed genes (DEGs) between SD0 and SD3 in those participants where vaccine shedding was evident in at least one time point ("Shedders", $n = 20$). **b** Biological processes enriched in DEGs identified in **a**. **c** Volcano plot of DEGs between SD0 and SD3 in those participants where vaccine shedding was not detected at any time point ("Non-Shedders", $n = 5$). **d** Biological processes enriched in DEGs identified in **c**. Levels of key DEGs in Shedders and Non-Shedders at SD0 and SD3; **e** *CXCL9*, **f** *CXCL10*, **g** *SOCS1*, **h** *CXCR1*. Median nasal cytokine kinetics in Shedders and Non-shedders over the study time course for **i** CXCL10, **j** CXCL8, and **k** IL-33 proteins. Panels **a**–**d** P-values attained by the Wald test were corrected for multiple testing using the Benjamini and Hochberg method. Panels **e**–**k** show median and interquartile range values.

As Fc-mediated antibody functions are increasingly recognized as being desirable for broadly reactive heterosubtypic humoral immunity to influenza[25], we conducted antibody dependent phagocytosis (ADP) assays with serum (Fig. 4e) and nasosorption eluates (Fig. 4f). The strongest induction was evident in serum against B/Vic ($P = 0.005$, Fig. 4e). Increases in phagocytic scores were also evident against these antigens with nasosorption eluates but did not reach statistical significance (Fig. 4f), likely owing to the relatively weak IgG response in nasal samples (Supplementary Fig. 3).

We next sought to determine whether serum IgG and nasal IgA responses were coordinated. Hierarchical clustering of serum IgG and nasal IgA fold-changes between SD0 and SD28 identified 3 clusters of responses amongst the cohort (Fig. 4g). The first cluster comprised 16/40 (40%) participants with minimal serum IgG responses but strong nasal IgA responses against a range of HA-antigens ("Naso-Ab"). The second cluster of participants had either no humoral response to LAIV, or lower-level responses evident in either the serum or nasal mucosa ("Mix-Ab", $n = 12$, 30%). The remaining 12 participants (30%) had predominantly serum-based antibody responses to LAIV, with little change in nasal IgA titre against any HA-antigen ("Sero-Ab"). Correlation matrix analysis showed that changes in antibody titre within a compartment were generally closely correlated whereas no correlation was seen between serum and nasal antibody responses to LAIV (Fig. 4h). Normalization of antigen-specific antibody titres to total isotype levels demonstrated that influenza-specific antibodies were more abundant in nasal samples than peripheral blood (Supplementary Fig. 6). Nasal IgA levels were equivalent between Naso-Ab, Mix-Ab and Sero-Ab groups at baseline (Supplementary Fig. 7). Serum IgG anti-B/Vic was significantly lower in the Sero-Ab group relative to Mix-Ab, but not Naso-Ab (Supplementary Fig. 7), indicating that serum antibody responses may be more common in those participants with lower baseline levels. Together, these data demonstrated that broad titre increases are seen in both nasal and serum antibody responses to LAIV, but that responses are strongly compartmentalized. Monitoring antibody responses in the nasal mucosa can therefore identify responses to LAIV that are not evident in peripheral blood.

## Peripheral blood lymphocyte responses to LAIV

We next sought to characterize the peripheral cellular response to LAIV, and associate this with the compartmentalized antibody responses evident between donors. PBMC samples were collected on SD0, SD7 and SD28 from 23 participants and analyzed by flow cytometry (gating strategy Supplementary Fig. 8). Total CD4+ T cell and major phenotype frequencies (Naïve (CD45RA+ CCR7+), T central memory ($T_{CM}$; CD45RA- CCR7+), T effector ($T_{EFF}$; CD45RA+ CCR7-), and T effector memory ($T_{EM}$; CD45RA- CCR7-) were stable between study timepoints (Fig. 5a, Supplementary Fig. 9A). The total CD8+ T cell frequency was similarly consistent, but the CD8+ $T_{EFF}$ population was expanded on SD7, alongside a contraction of the naïve CD8 pool (Fig. 5b, Supplementary Fig. 9B). Fold-change analysis between SD0 and SD7 for each donor confirmed that the total Naïve, and $T_{CM}$, CD8 T cell pools contracted at SD7 while the CD8+ $T_{EFF}$ frequency expanded (Fig. 5c). To study whether this expanded total CD8 $T_{EFF}$ population was trafficking to the mucosa we analyzed the change in CXCR3 expression by these cells, indicating that the CXCR3+ population of CD8+ $T_{EFF}$ increases in some participants at SD7 but contracts in others

(Fig. 5d). The frequency and activation of circulating T follicular helper cells (cTfh; CD4+ CXCR5+ T cells) and CD19+ B cells and antibody secreting cells (ASCs; CD27+ CD38+ B cells) was also studied (Fig. 5e, Supplementary Fig. 9C). Activation of cTfh was analyzed as the frequency of ICOS+ PD-1+ cells within the total cTfh, cTfh1-like (CXCR3+), and cTfh2/17-like (CXCR3-) populations, demonstrating that activation relative to SD0 was evident within the cTfh1-like population at SD7 in a subset of participants. In line with a recent report of cTfh responses to intramuscular influenza vaccine[26], we also observed robust increases in the frequency of ICOS+ CD38+ cTfh at SD7, relative to both SD0 and SD28 (Fig. 5g, both $P < 0.05$).

To determine whether the heterogenous activation of CD8+ T cells (expansion of the CD8+ $T_{EFF}$ population), cTfh activation, and increases in ASCs frequency were coordinated, a correlation matrix analysis of population fold-changes between SD0 and SD7 was performed (Fig. 5h). This demonstrated that the scale of CD8+ $T_{EFF}$ expansion was associated with the contraction of the naïve CD8+ pool, but these CD8 responses were not associated with cTfh or B cell responses. While classical cTfh activation (ICOS+ PD-1+) was not associated with either CD8+ responses or ASC frequency, a significant positive correlation was evident between ICOS+ CD38+ cTfh and ASCs (Fig. 5h). We next sought to determine the association between these peripheral blood cellular responses to LAIV at SD7 and the serum IgG and nasal IgA responses at SD28, focusing on B/Yam as the antigen against which responses were most common. The frequencies of both ICOS+ CD38+ total cTfh and ICOS+ PD-1+ cTfh1-like cells at SD7 significantly positively correlated with serum IgG responses at SD28 (Fig. 5i), but not nasal IgA (Supplementary Fig. 9D). By contrast, the expansion of the CD8+ $T_{EFF}$ population at SD7, relative to SD0, and the concomitant contraction of the CD8+ naïve population, were both significantly associated with nasal IgA responses at SD28 (Fig. 5j), but not serum IgG responses (Supplementary Fig. 9E). Together, these data indicated that activation of peripheral blood cTfh and B cells at 7 days post-LAIV are associated with serum IgG responses. By comparison these responses are not associated with the mucosal IgA response, which is instead associated with the activation of peripheral CD8+ T cells.

## Distinct early mucosal immune responses to LAIV between nasal and serum antibody response groups

Finally, we sought to understand whether the mucosal immune responses in the first 72 h p.i. were associated with these compartmentalized antibody responses to LAIV. The Naso-Ab, Mix-Ab, and Sero-Ab antibody response groups identified in clustering analysis (Fig. 4g) had equivalent mucosal antibody titres at baseline (Supplementary Fig. 7), but were associated with different profiles of vaccine shedding. Shedding of both B/Yam and B/Vic at 24 h p.i. was more common in participants that developed serum antibody responses at SD28 (Fig. 6a, 6b, respectively), though the extent of shedding was not significantly different between groups. This was in agreement with the seroconversion evident in some children after LAIV vaccination, despite a lack of vaccine virus shedding[20]. Similarly, mucosal inoculation, but not viral shedding, has been reported to be essential for LAIV immunogenicity in mice[27]. At SD3 there were 80 upregulated DEGs between the Naso-Ab and Sero-Ab antibody response groups (Fig. 6c). BP analysis of the DEGs identified pathways associated with antiviral immunity (including "Antiviral innate response" and "T cell

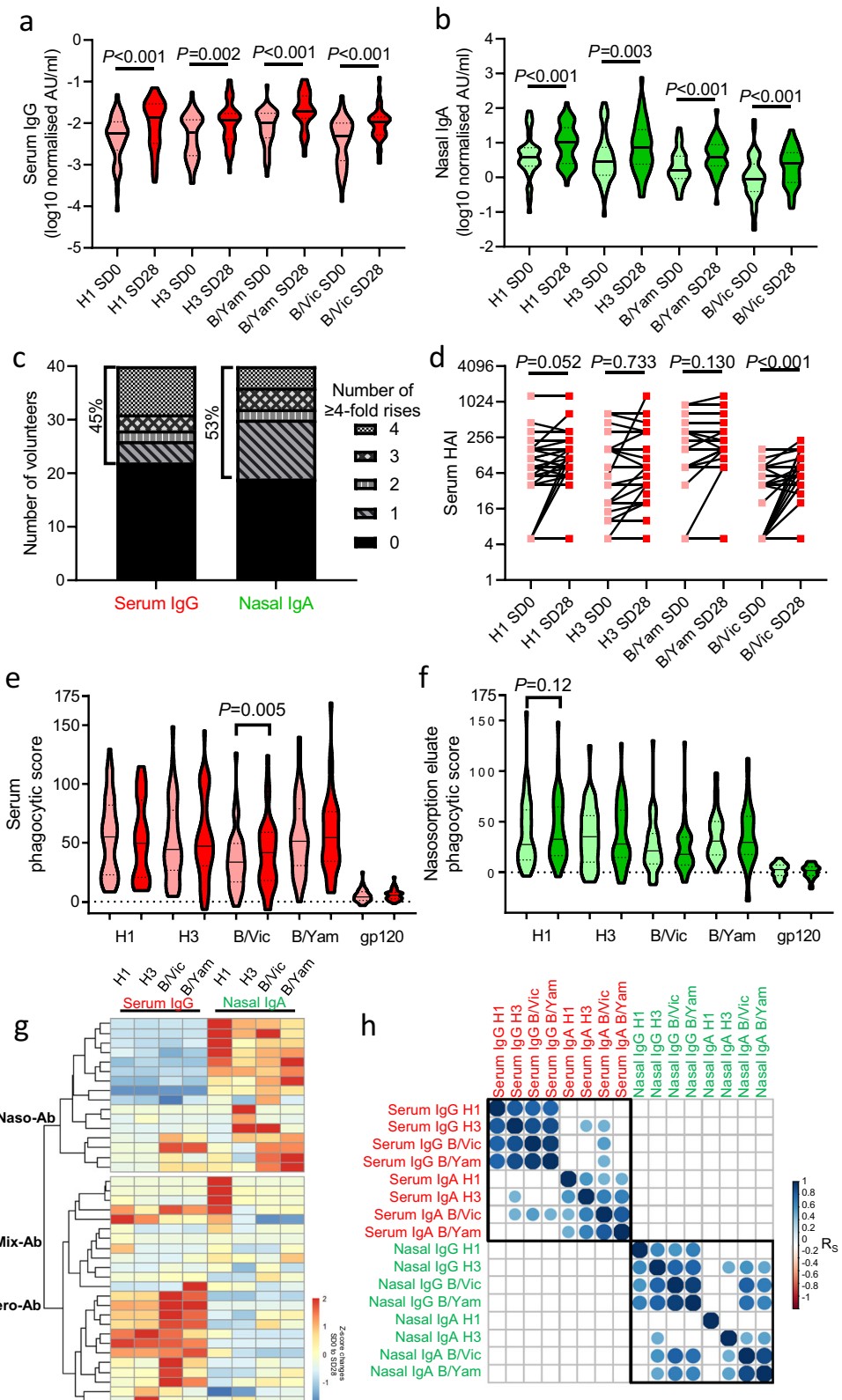

chemotaxis") that were elevated at SD3 in the Naso-Ab group, relative to Sero-Ab (Fig. 6d), in line with the better control of viral shedding evident in the Naso-Ab group (Fig. 6a, b). We next determined whether these differences in antiviral immunity at SD3 between antibody response groups were due to baseline differences or differences in induction following LAIV inoculation. Analysis of key DEGs at SD0 and SD3 in both groups showed baseline (SD0) levels of *CXCL10*, *DDX58*,

*OASL*, and *LY9* were equivalent between groups (Fig. 6e–h, respectively). *CXCL10* was induced following LAIV in all groups (Fig. 6e), while the potent viral suppressors *DDX58* and *OASL* were induced only in the Naso-Ab group (Fig. 6f, 6g, respectively). Suppression of baseline *LY9* (a SLAM family member immunomodulatory receptor expressed by some T cell populations[28]) was evident at SD3 in the Naso-Ab group only (Fig. 6h). In agreement with this enhanced IFN-driven immune

**Fig. 4 | Antibody responses to LAIV in the blood and nasal mucosa are independent.** Multiplex immunoassays were utilised to measure antigen-specific antibody titres against each LAIV haemagglutinin (HA) at SD0 and SD28 in participants ($n = 40$); **a** Serum IgG, **b** Nasal IgA. **c** The frequency of conversions ($\geq$4-fold titre rises) in serum IgG and nasal IgA between SD0 and SD28. **d** Serum HA inhibition (HAI) assay titres at SD0 and SD28 for each vaccine HA. Antibody dependent phagocytosis assays results using opsonised HA coated beads (or HIV gp120 as negative control) for **e** serum and **f** nasal fluid at SD0 (light colors) and SD28 (dark colors). **g** Hierarchically clustered heatmap of Z-scored serum IgG and nasal IgA changes between SD0 and SD28 for each participant for all 4 vaccine HA antigens.

Participant clusters are labelled with their predominant humoral response; largely nasal IgA ("Naso-Ab", $n = 16$), largely serum IgG ("Sero-Ab", $n = 12$), or a mixed/absent humoral response ("Mix-Ab", $n = 12$). **h** Correlation matrix of nasal and serum IgG and IgA changes between SD0 and SD28 in all participants. Significance was tested between groups using two-sided Wilcoxon tests or Mann-Whitney $U$ tests. Correlations used Spearman R-values, where blank spaces denote non-significant correlations after $P$-value adjustment for multiple testing. Panels **a**, **b** represent antibody binding titre data in arbitrary units (AU). Lines and dotted lines in **a**, **b**, **e**, and **f** show medians and interquartile ranges, respectively. Source data are provided as a Source Data file.

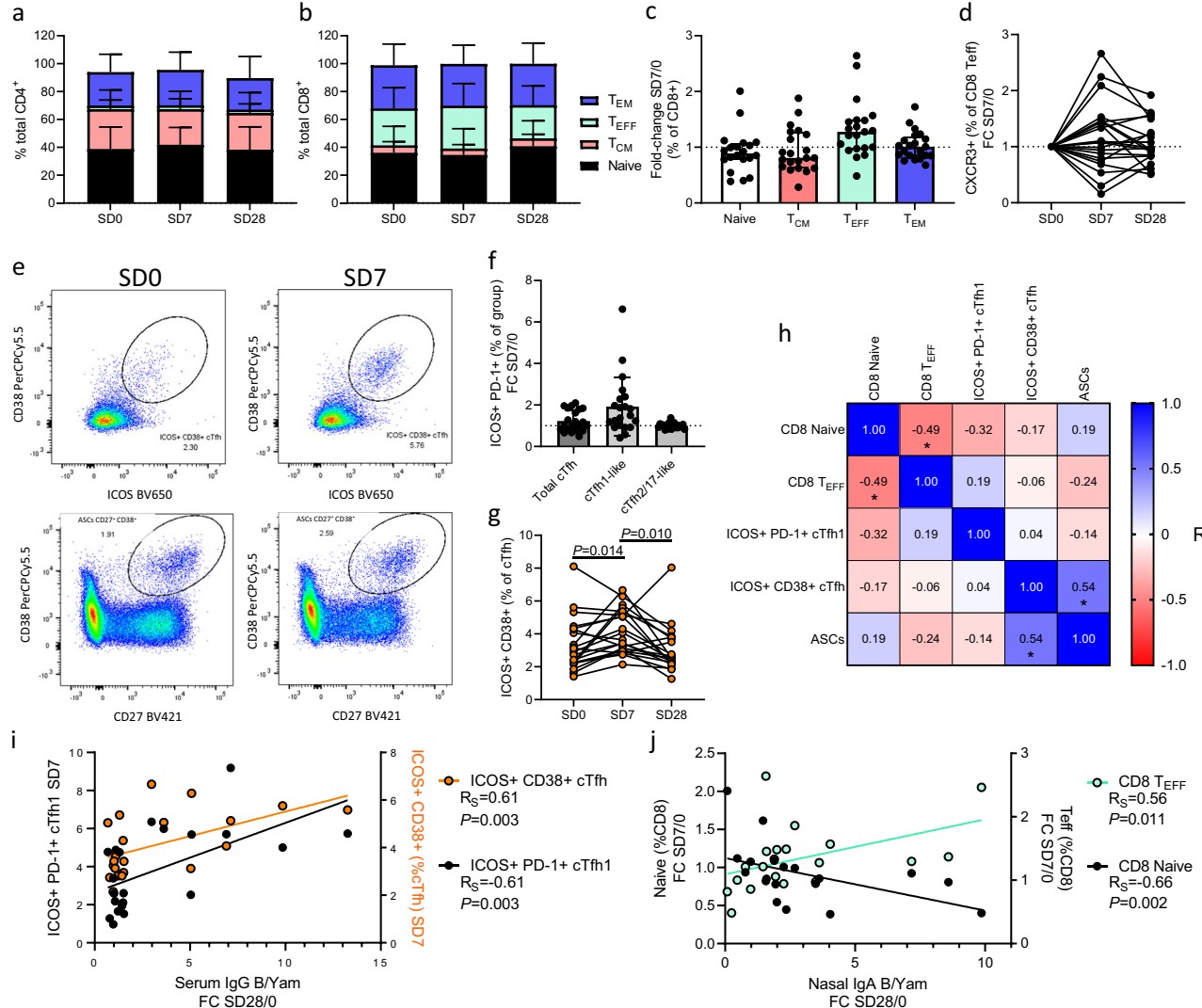

**Fig. 5 | Peripheral blood lymphocyte activation following live attenuated influenza vaccination.** Phenotypic analysis of peripheral blood mononuclear cells (PBMCs) following live attenuated influenza vaccine (LAIV) inoculation of young adults ($n = 23$). **a** CD4$^+$ T cell and **b** CD8$^+$ T cell subtype frequencies at SD0, SD7, and SD28 based on expression of CCR7 and CD45RA ($T_{EM}$ = T effector memory, $T_{EFF}$ = T effector, $T_{CM}$ = T central memory). **c** CD8$^+$ subpopulation frequency fold-changes between SD7 and SD0. **d** Fold-change of CXCR3$^+$ CD8$^+$ $T_{EFF}$ frequency at SD7 and SD28, relative to SD0. **e** Activated circulating T follicular helper (cTfh) (ICOS$^+$ CD38$^+$ of total CD4$^+$ CXCR5$^+$ T cells) and antibody secreting cells (ASCs; CD27$^+$ CD38$^+$ of total CD19$^+$ B cells) frequencies at SD0 and SD7. **f** Fold-change between SD7 and SD0 in the frequency of activated (ICOS$^+$ PD-1$^+$) total cTfh and cTfh1-like (CXCR3$^+$) and cTfh-2/17-like (CXCR3$^-$) subpopulations (dashed line denotes fold-change of 1, indicating no change between SD7 and SD0). **g** Frequency over time of alternatively

activated (ICOS$^+$ CD38$^+$) cTfh. **h** Correlation matrix of key PBMC changes between SD7 and SD0. **i** Correlation between serum IgG anti-B/Yamagata titre fold-changes between SD28 and SD0 versus cTfh1-like cell activation (ICOS$^+$ PD-1$^+$) and alternatively activated (ICOS$^+$ CD38$^+$) total cTfh at SD7. **j** Correlation between nasal IgA anti-B/Yamagata titre fold-changes between SD28 and SD0 versus naïve and $T_{EFF}$ CD8$^+$ T cell fold-changes at SD7 relative to SD0. Panels **a** and **b** represent mean + standard deviation frequencies. Panels **c** and **f** represents median and interquartile ranges. Statistical testing in **g** used two-sided ANOVA with correction for multiple testing. Panel **h** assessed correlation using two-sided Pearson's methods for parametrically distributed data where numbers denote Pearson R values and asterisks denote significant correlations. Panels **i** and **j** utilized two-sided Spearman's correlations for testing non-parametrically distributed data. *$P < 0.05$, **$P < 0.01$. Source data are provided as a Source Data file.

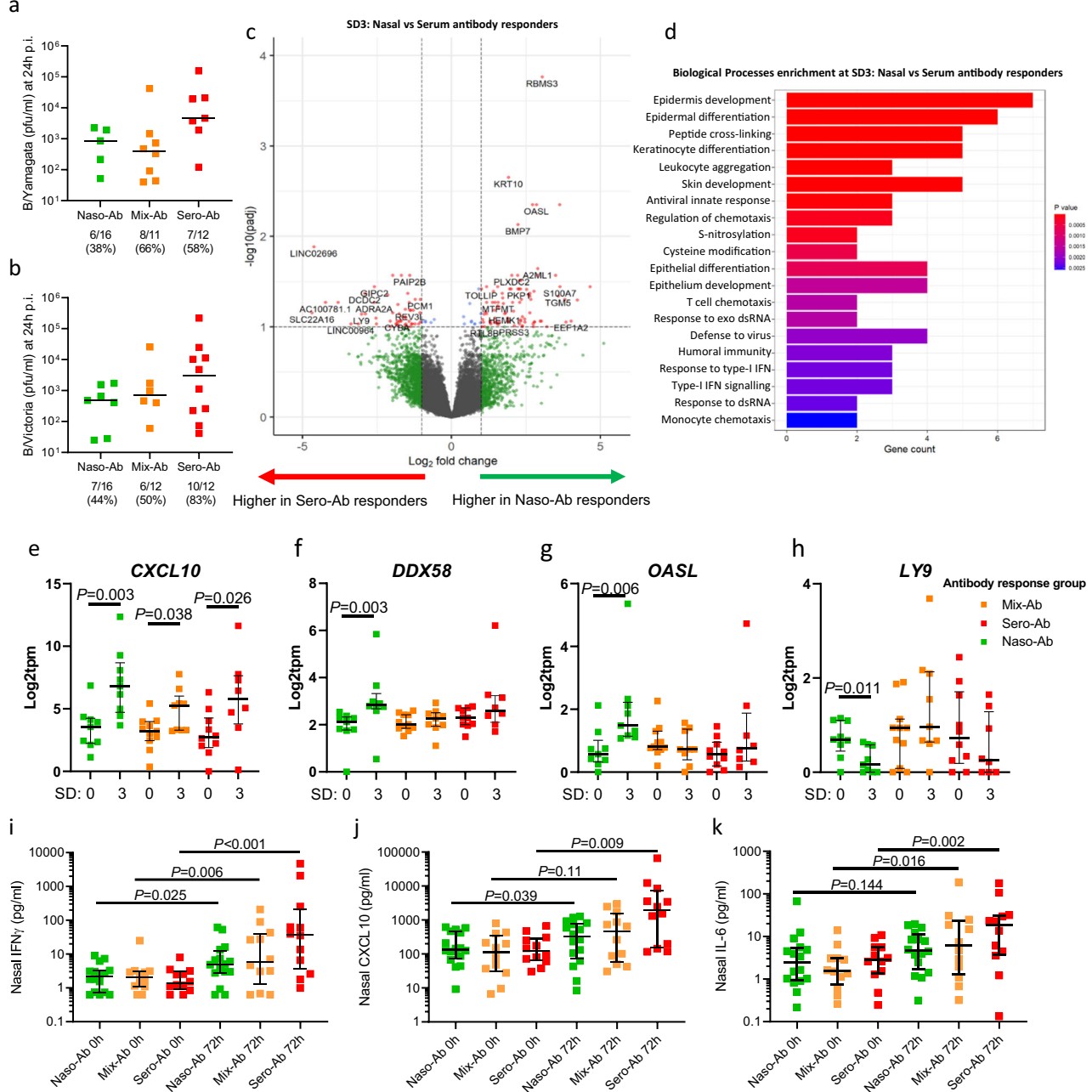

**Fig. 6 | Divergent mucosal responses to LAIV between nasal and blood antibody response groups.** Participants ($n = 40$) were segregated into groups of predominantly nasal ("Naso-Ab", $n = 16$), blood ("Sero-Ab", $n = 12$) or mixed ("Mix-Ab", $n = 12$) antibody responses to LAIV, based on clustering in Fig. 3g. Viral load between Naso-Ab, Mix-Ab and Sero-Ab groups for **a** B/Yamagata and **b** B/Victoria. **c** Differentially expressed genes (DEGs) in nasal tissue samples at SD3 between Naso-Ab and Sero-Ab response groups. **d** Biological processes enriched in the DEGs from **c**. Expression of **e** *CXCL10*, **f** *DDX58*, **g** *OASL*, and **h** *Ly9* at SD0 and SD3 in the Mix-Ab, Naso-Ab, and Sero-Ab response groups. Nasal protein levels of **i** IFNγ, **j** CXCL10, and **k** IL-6 at baseline (0 h) and SD3 (72 h) post-vaccination. Panels **e**–**k** represent medians and interquartile ranges. Panels **e**–**h** represent *P*-values attained by the Wald test corrected for multiple testing using the Benjamini and Hochberg method. In **i**–**k**, significance was tested between baseline (0 h) and 72 h post-inoculation within each group using Wilcoxon tests.

response in the nasal mucosa at SD3 in the Sero-Ab group, protein levels of IFNγ, CXCL10, and IL-6 were greatest in the Sero-Ab group at 72 h p.i., though induction of these mediators was still evident in the Naso-Ab and Mix-Ab groups (Fig. 6e–g, respectively). While IL-33 induction at 8 h p.i. was equivalent between antibody response groups, the inverse association between IL-33 release and the subsequent scale of vaccine shedding (Fig. 2h–i) may indicate that early alarmin release suppresses viral replication via mechanisms independent on the subsequent IFN-led mucosal anti-viral immune responses that in turn

associate with distinct mucosal and peripheral antibody responses to LAIV in healthy young adults (Fig. 7).

## Discussion

While an early IL-33 response associated with suppression of viral replication may be desirable during viral infection, during vaccination this alarmin was associated with decreased viral replication and consequential mucosal IFN responses, limiting peripheral antibody responses. However, despite decreased viral replication and mucosal

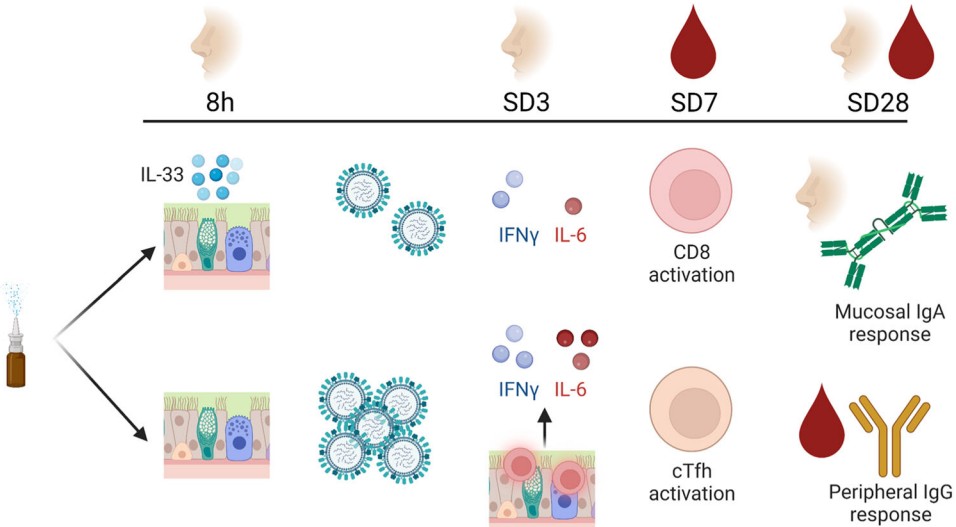

**Fig. 7 | Schematic summary of mucosal and systemic immune responses to LAIV.** Following inoculation with live attenuated influenza vaccine (LAIV), the earliest observed mucosal immune response was an elevation of IL-33 within 5-8 h post-inoculation. The scale of this IL-33 release was inversely associated with the scale of LAIV replication in the nose, which was observed in ~50% of volunteers for each of the Influenza B virus constituents. By 72 h (SD3) post-inoculation, the transcriptional response to LAIV in the airway is dominated by interferon responses, also observed as protein level increases. Transcription of interferon stimulated genes like *OASL* and *DDX58* is greatest in the "Naso-Ab" group while protein levels of

IL-6 are highest in the 'Sero-Ab' group at SD3. At 7 days post inoculation (SD7), CD4 and CD8 T cell activation is evident, alongside activation of B cell populations. CD8 T cell activation appeared distinct from T follicular helper cell (cTfh) and B cell activation. The scale of these lymphocyte responses was in turn associated with divergent mucosal IgA and peripheral blood IgG responses to LAIV antigens. Mucosal IgA (akin to the Naso-Ab group) responses were stronger in participants that typically had higher CD8 T cell activation, while peripheral blood IgG responses (akin to the Sero-Ab group) were associated with cTfh and B cell activation. Created with Biorender.com.

IFN responses, many participants that did not mount blood antibody responses did develop raised levels of mucosal virus-specific IgA.

These results are potentially relevant to the development and immunogenicity testing of live attenuated vaccines for respiratory viruses including influenza, respiratory syncytial virus (RSV), and SARS-CoV-2. First, immunogenicity testing that relies only on peripheral blood antibody responses is likely to miss relevant mucosal antibody responses. We found these mucosal and blood antibody responses to LAIV to be distinct, indicating a strong degree of compartmentalization in agreement with the recent demonstration of a segregated mucosal plasma cell populations in the upper airway[29]. Mucosal antibodies are considered particularly important in providing sterilizing immunity against respiratory infections[30], while circulating antibodies may be more important for attenuating disease severity and protecting the lower respiratory tract once an infection occurs[2]. The present results demonstrate that such mucosal antibody responses can be mediated independently of peripheral antibody responses. We demonstrate that these distinct mucosal and peripheral antibody responses are associated with differences in early mucosal alarmin and IFN-led anti-viral responses and vaccine replication. Manipulation of these pathways may favor the induction of mucosal or systemic antibodies after vaccination with live attenuated viruses.

Replication of LAIV, evidenced by viral shedding, has been considered an important determinant of vaccine efficacy[7,23]. However, an increasing weight of literature supports the principle that detectable viral shedding may not be a pre-requisite for the induction of protective immunity[20,31]. We demonstrate that in young adults, shedding of either H1 or H3 constituents was rarely evident but vaccination still induced robust antibody responses, including serum HAI seroconversions, in some participants. One analysis of humoral responses of adults to LAIV noted that 9% of participants had >4-fold serum HAI titre rises to H1N1 and 24% of participants had increased binding antibodies against H1N1, measured by immunohistochemistry[18]. These results closely align with responses against H1N1 we observed, where 4/40 (10%) of our cohort had serum HAI responses and 8/40 (20%) had >4-fold rises in serum IgG to H1N1. Viral shedding in our study was

associated with some differences in the nasal mucosal transcriptional profile at baseline, though these DEGs did not associate with any biological or immunological processes in a pathways analysis. Reduction in *LY9* expression at SD3, relative to SD0, in the Naso-Ab group only may represent a transient suppression of tolerance that facilitates mucosal antibody responses, as loss of Ly9 has been associated with a lack of tolerance and formation of autoantibodies[28]. Others have reported that lower LAIV shedding is associated with mucosal IFN responses at baseline in children, in part owing to high rates of asymptomatic respiratory virus carriage[31]. The small number of Non-Shedders in our analysis, and lower viral carriage rates generally observed amongst adults[32], may have resulted in our inability to reproduce this result. Additionally, respiratory virus carriage in children has been reported to not influence LAIV immunogenicity[8], supporting the concept that vaccine replication has only modest influence on humoral immune responses. One fifth of participants in our study were Non-Shedders and further studies of immune responses in this group are warranted. Assuming consistency in the ratio of Shedders to Non-Shedders, a study size of 127 participants would be required to observe significant differences in nasal CXCL10 levels at 72 hours post-inoculation.

We utilized nasosorption[33] for studying mucosal antibody responses, in addition to mucosal cytokine and chemokine levels following LAIV inoculation. Nasosorption has been reported to yield more concentrated antibody than alternative upper airway sampling techniques[34] and is well tolerated by participants[35]. This concentration of mucosal antibody enabled us to utilize an antibody-dependent phagocytosis assay of mucosal antibody function, opening the possibility of functional assessments of mucosal antibodies. Emerging evidence highlights the importance of antibody effector functions in protection against RSV[36] and influenza[37]. Further development of assays for neutralization and Fc-mediated effector functions of mucosal antibodies is therefore crucial for our understanding of compartmentalized antibody functions.

Our data are limited by the lack of long-term follow-up of our cohort following vaccination. Such follow-up would enable the

determination of the duration of mucosal antibody responses following LAIV, but are confounded by community exposure to influenza viruses. Furthermore, our results in young adults cannot be assumed to apply to children or older adults. Seroconversion evident in the blood, alongside peripheral cTfh and B cell activation, may be associated with the generation of long-lived responses[38]. By comparison, the duration of compartmentalized mucosal IgA responses that may lack long-lived peripheral plasma cell support is uncertain. While our study design enabled us to observe IL-33 in the first 8 h p.i., and the inverse association of this response with the scale of subsequent vaccine shedding, the mechanism that links IL-33 to viral suppression requires further study. IL-33 is an initiator of type-2 immune responses through IL-1RL1 (ST2) dependent activation of ILC2 cells[21] but we saw no evidence of type-2 responses post-LAIV. This suggests that a type-2 independent mechanism may be active. IL-1RL1 expression can be induced on a diverse range of cells, including natural killer cells, resulting in sensitivity to IL-33[39]. Alternatively, oxidized-IL-33 has recently been proposed to signal through RAGE/EGFR which may trigger alternative host defense responses including mucus production[40]. IL-33 induction was not associated with the prototypical IFN-led anti-viral response that dominated the mucosal response to LAIV at SD1 or SD3, nor was IL-33 directly associated with antibody responses. However, our data indicates that early IL-33 release is associated with a mucosal response that limits viral replication, the scale of which determines the scale of mucosal IFN responses and hence the type of antibody response that is initiated. IL-33 can directly activate CD8+ T cells[41], which was recently shown to favor retention of stem-like properties over progression to a terminally differentiated state[42]. LAIV triggers a CD8+ T cell response against diverse vaccine antigens not seen in response to other vaccine types[43]. Our observation of peripheral CD8+ T cell and cTfh activation therefore likely represents both an activation and expansion of diverse antigen-specific cells, and a degree of bystander T cell activation. This activation of CD8+ T cells in participants forming nasal antibody responses may also indicate a stronger generation of cellular immunity that has been implicated in expedited viral clearance in animal models[44]. The activation of CD8+ T cells by IL-33 concords with our observation of peripheral CD8+ T cell activation being associated with mucosal antibody responses. In support of this association, nasally administered IL-33 has been reported to enhance the induction of mucosal IgA following inactivated influenza vaccination in mice[45]. The small-scale early IL-33 response in our study likely triggers distinct responses from the IL-33 release seen during severe respiratory viral infection[46] that has been associated with poor patient outcomes[47]. It will be of interest to distinguish these differential roles for IL-33 in anti-viral immunity in healthy individuals and patients with severe viral infections.

A correlate of efficacy for LAIV has been elusive, the weak antibody response in the blood failing to reflect the protective effect of vaccination. Our data indicate that the efficacy of LAIV may result from antibody responses confined to the nasal mucosa, which may provide protection not reflected by blood-based studies.

## Methods

### Recruitment and ethics
The study was approved by the London Camberwell St Giles Research Ethics Committee (REC, reference 18/LO/0904) and the Health Research Authority. The study was registered on clinicaltrials.gov (NCT04110366). Participants were recruited by local advertisement and written informed consent was obtained. Exclusion criteria were age <18 years or >30 years, receipt of an influenza vaccine in the last 2 years, egg allergies, current smoking, pregnancy, use of any medication that may affect the immune system (including anti-histamines), current acute illness (including respiratory infections), clinically diagnosed influenza in the last 2 years, any long-term health problem (including asthma and other conditions that would trigger a

recommendation for influenza immunization), history of Guillain-Barre syndrome, receipt of any vaccine in the past 4 weeks, and living with an immunocompromised person.

The pre-defined primary outcome of the study was the frequency of nasal vaccine virus shedding between days 1-7 post-inoculation of LAIV. The pre-defined secondary outcome measure was the frequency of >2-fold rises in mucosal and/or serum antibody titres against the haemagglutinin antigens of each LAIV constituent virus at 28 days post-inoculation.

### Study schedule and clinical sampling
Prospective study participants were invited to a screening visit at which consent to participate was sought. Baseline samples were collected at the screening visit following written informed consent. Participants were not serologically prescreened for enrolment to the study. Participants returned to the research unit ~7 days after screening for the nasal vaccination visit. The vaccination visit entailed inoculation with Fluenz Tetra (AstraZeneca) followed by nasosorption sampling[33] at 3, 4, 5, 6, 7, and 8 h post-inoculation. Participants then returned to the clinic for follow-up visits 1, 3, 7, and 28 days after vaccination. At each visit total nasal symptom score (TNSS) and peak nasal inspiratory flow (PNIF) measurements were taken, as described elsewhere[48], and symptoms solicited. Vehicle control study participants undertook a single study visit lasting 8 hours.

Nasosorption samples were collected as previously described[33] with a 1 minute absorption duration. Nasosorption samples were eluted immediately with 300 μl of either Millipore immunoassay buffer AB-33k (Merck) or Qiagen RLT lysis buffer. Nasal curettage was performed with Rhino-Pro curettes (Arlington Scientific) which were collected into RNA Shield (Zymo Research) and immediately frozen at −80 °C. Peripheral blood was collected into either EDTA or Heparin tubes, for the isolation of peripheral blood mononuclear cells or serum.

### Nasal viral load quantification
Nasal viral RNA was extracted from nasosorption samples that had been eluted in RLT lysis buffer using a MagMax viral RNA isolation kit (ThermoFisher Scientific, #AM1939) and quantified at UKHSA using an LAIV-specific multiplex RT-qPCR assay as previously described[20]. Briefly, extracted nasal viral RNA samples were analyzed alongside RNA standards consisting of viral RNA extracted from known plaque forming unit (PFU) quantities of four recombinant LAIV vaccine viruses: an A/Bolivia/559/2013 (H1N1)pdm09-like virus, an A/Switzerland/971593/2013 H3N2-like virus, a B/Phuket/3073/2013-like B/Yamagata-lineage virus and a B/Brisbane/60/2008-like B/Victoria-lineage virus. RNA standards were used to generate a standard curve of PFU vs Ct, which was then used to determine the quantity in PFU/ml of the four viruses in each nasal test sample based on their Ct values. For each participant PFU values for samples taken on days 1, 3 and 7 post-inoculation were log-transformed and area under the curve (AUC) titre calculated in PFU/ml per day.

### Mucosal mediators
Levels of IFN-α2a/β/γ/λ (IL-29), IL-1β, IL-2, IL-4, IL-6, IL-8, IL-10, IL-12p70, IL-13, TNFα, BAFF, IL-21, IL-33, and CXCL10 (IP-10) were quantified from nasosorption eluates using Mesoscale Diagnostics (MSD) V-Plex and U-Plex multiplex immunoassay plates as per manufacturer's protocol. Where results for an individual mediator are not covered in the results, the mediator was either below the lower limit of quantitation, or was not significantly altered following LAIV challenge at any time point. All MSD plates were read on an MSD QuickPlex SQ120 instrument using Mesoscale Discovery Workbench software (v4.0, MSD).

### Antibody titre quantification
IgG and IgA titres against recombinant influenza hemagglutinin antigens were determined using a custom MSD multiplex immunoassay.

Individual spots were coated with either H1N1 A/Slovenia2903/2015 (Immune Tech, #IT-003000107ΔTMp), H3N2 A/Switzerland/9715293/2013 (Protein Sciences, #3006_H3_Sw), B/Brisbane/60/2008 (Astra-Zeneca, "B/Victoria"), or B/Phuket/3073/2013 (AstraZeneca, "B/Yama-gata"). Antibody titres within samples were determined relative to an arbitrary unit (AU) scale, which was generated on an 8-point standard curve using a pooled human serum sample (AstraZeneca). Inter-plate consistency was confirmed using two blended human nasal fluid samples and three human serum control samples (AstraZeneca). Detection was performed using MSD Sulfo-Tag conjugated anti-human IgA (#D20JJ-6) or IgG (#D20JL-6) antibodies. Specific antibody binding titre results were normalized to the total isotype (IgG or IgA, as appropriate) content in each sample using Human ProcartaPlex iso-typing multiplex panels (ThermoFisher Scientific) with a Bio-Plex 200 instrument and Bio-Plex Manager software (v6.0)(both Bio-Rad).

### Antibody dependent phagocytosis assay

Antibody dependent phagocytosis assays were performed according to an amended protocol detailed elsewhere[49]. Briefly, NeutrAvidin microspheres (Invitrogen) were labelled with either an aforementioned recombinant influenza hemagglutinin or HIV-1 gp120 protein (Sinobiological, #11233-V08H) that had been biotinylated using EZ-Link Sulfo-NHS-SS biotinylation kits (ThermoFisher Scientific, #21945). After twice washing in PBS-1% BSA, beads were reconstituted at 1:100 in PBS-1%BSA. $9 \times 10^5$ beads were incubated with antibody samples for 2 h at 37 °C, washed in PBS-1%BSA and $2 \times 10^4$ THP-1 human monocytic cells were added for 18 h. Cells were then fixed in 2% paraformaldehyde and acquired on a Fortessa cytometer equipped with a high-throughput system plate reader. A minimum of 2000 cell events were captured. Phagocytic score was determined as the % of cells bead positive multiplied by the mean fluorescent intensity of the bead positive region. Assays were performed in technical triplicate. All assays for a given antigen were performed using a single batch of antigen-coated beads.

### Hemagglutination inhibition assay

Sera were analyzed to ISO 15189 standards by Hemagglutination Inhibition (HAI) with A/Michigan/45/2015 (H1N1)pdm09, A/Singapore/INFIMH-16-0019/2016 (H3N2), B/Colorado/06/2017 and B/Phuket/3073/2013. HAI was performed by UK Health Security Agency using methods previously described[50] with a modification to the serum treatment whereby all sera were treated with Receptor Destroying Enzyme from Vibrio cholerae (RDE II, Denka Seiken) according to manufacturer's recommendation and haem-adsorbed using turkey erythrocytes prior to analysis.

### Transcriptomics

Nasal curettage samples stored in RNAShield had total RNA extracted using Lexogen Split RNA extraction kits (Lexogen, #008). Total RNA samples were sent to Lexogen for library preparation using QuantSeq FWD kits (Lexogen, #015) and RNA sequencing using an Illumina NextSeq 500 in SR75 mode. Fastq files were generated from a total of 70 curettages samples. Sequencing quality control was performed using FastQC (v0.11.7)[51], samtools stats (v1.9)[52] and Qualimap (v2.2.2c)[53]. Quality control (QC) metrics obtained with Qualimap were based on alignment performed with STAR[54] against human reference genome GRCh38 Ensembl 99. MultiQC(1.9)[55] was used to summarize and asses the libraries QC metrics for sample selection. From these analysis, 10 samples were removed due to low read quality, leaving 60 samples for further analysis. Next, Fastp (v 0.20.1)[56] was used to trim the adapters and reads were mapped against Ensembl 99 and quantified using the Salmon tool[57]. Nextflow workflow[58] and Bioconda software management tool[59] was used to run the workflow.

DeSEQ2 (v 1.34.0) was used to normalize the counts and to identify differentially expressed genes (DEG) between groups in R (v

4.0.0). In DESeq2, the *P*-values attained by the Wald test were corrected for multiple testing using the Benjamini and Hochberg method. For the DEG analysis, genes displaying <5 counts in >50% of the samples used for the comparisons were filtered out. We identified differentially expressed genes up or down regulated with log2 fold-change ≥1 or ≤ −1 and adjusted *P*-value < 0.1. For visualization, volcano plots were generated using R package EnhancedVolcano R package (v 1.8.0) and ggplot2 (v 3.3.5). For gene ontology analysis the list of DEGs obtained from DeSEQ1 were applied as input and performed using EnrichR[60] "GO Biological Process". Transcriptomics data were deposited in GEO with the accession identifier GSE230494.

### PBMC isolation and flow cytometry

Peripheral blood mononuclear cells were isolated using histopaque gradients and frozen ahead of cytometry. The flow cytometry antibody staining cocktail comprised antibodies from BD: anti-CD4 (clone SK3, BUV496, Cat#564651, Lot#6259653, 1:20), -CD8 (clone RPA-T8, BUV395, Cat#563795, Lot#7069910, 1:40), and -CD278 (Inducible T-cell COStimulator, ICOS; clone DX29, BV650, Cat#563832, Lot#7198562, 1:20) and all remaining antibodies were from Biolegend: anti-CD19 (clone HIB19, BV510, Cat#302242, Lot#B239285, 1:20), -CD279 (Programmed cell death protein 1, PD-1; clone EH12.2H7, BV605, Cat#329924, lot#B238715, 1:20), -CCR7 (clone G043-H7, PE/Cy7, Cat#353226, Lot#B238508, 1:20), -CD38 (clone HB-7, PerCPCy5.5, Cat#356613, Lot#B239525, 1:40), -CD32 (clone FUN-2, PE, Cat#303205, Lot#B211165, 1:80), -CD45RA (clone HI100, FITC, Cat#304106, Lot#B202186, 1:40), -CD27 (clone M-T271, BV421, Cat#356417, Lot#B243455, 1:40), -CD127 (clone A019D5, BV711, Cat#351327, Lot#B228247, 1:40), -CXCR3 (clone G025H7, BV786, Cat#353737, Lot#B236359, 1:40), -CD3 (clone OKT3, AF700, Cat#353737, Lot#B223632, 1:80), -CXCR5 (clone J25D4, PE/Dazzle, Cat#356928, Lot#B226046, 1:80) and live dead NIR dye (ThermoFisher Scientific, 1:1000). Fixation, staining, and acquisition were performed as previously described[61], with acquisition performed on a BD LSR Fortessa with BD FACSDiva software (v6.1.3., BD Biosciences). All flow cytometry data analysis was conducted using FlowJo v10.4 (FlowJo LLC).

### Statistical analysis

Statistical analyses were conducted using GraphPad Prism v9 or R v4.2.1. Data distribution was tested by D'Agostino and Pearson normality tests. Non-parametrically distributed were analyzed by ANOVA using Kruskal-Wallis tests with Dunn's test for multiple comparisons of repeat measures for 3+ groups. Non-parametric two-way analyses used Mann-Whitney *U* tests or Wilcoxon tests for unpaired or paired data, respectively. Correlation matrices used GraphPad Prism or R packages *ggplot2* and *ggcorrplot* and Spearman's test for correlation of non-parametric data, with *P*-value adjustment for multiple testing. Heatmaps of log10 transformed, scaled and centered antibody data were generated using the *ComplexHeatmap* package in R with rows and columns split by K-means clustering and dendrograms based on Ward's minimum variance method (ward.D2) and Spearman's rank correlations.

### Reporting summary

Further information on research design is available in the Nature Portfolio Reporting Summary linked to this article.

## Data availability

The transcriptomics data generated in this study have been deposited in GEO under the accession identifier GSE230494. The human GRCh38 reference genome assembly is available through the National Center for Biotechnology Information RefSeq assembly GCF_000001405.40. The remaining data that support the findings of this study are available from the corresponding authors upon reasonable request. Source data are provided with this paper.

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

## Acknowledgements

This study was funded by the National Institute for Health and Care Research (NIHR) Imperial Biomedical Research Centre (BRC; NIHR Imperial BRC grant P70668, R.S.T., T.T.H. and P.J.M.O.) and the Human Infection Challenge for Vaccine development (HIC-Vac, R.S.T. and P.J.M.O.) consortium (funded by the Wellcome Trust, UKRI Medical Research Council (MRC), and Global Challenges Research Fund (GCRF)). K.M.P. was supported by a St Mary's Development Trust Fellowship and the NIHR Imperial BRC and has received research grants from the Chan Zuckerberg Initiative, the NIHR, BRC, VTF, MRC/UKRI and a Fellowship from MRC/UKRI (MR/W024977/1). P.J.M.O. is supported by an NIHR Senior Investigator award. The study was supported by the Health Protection Research Unit (HPRU) in Respiratory Infections at Imperial College London in partnership with Public Health England/UK Health Security Agency and the NIHR Clinical Research Network (CRN). N.S. was supported by an AstraZeneca funded studentship. The views expressed are those of the authors and not necessarily those of the NIHR or the Department of Health and Social Care. We thank Mark Esser (AstraZeneca) for provision of multiplex immunoassay plates and standards for the quantification of anti-HA antibodies. AstraZeneca were not involved in the design or implementation of the study protocol.

## Author contributions

R.S.T., T.T.H., and P.J.M.O. conceived the study and sought ethical approvals. N.S. and N.P. recruited and consented participants, collected clinical specimens and maintained clinical records. R.S.T., A.S.S.U., and M.C. processed clinical specimens. K.M.P. designed and optimized flow cytometry experiments. D.J. conducted viral quantification experiments and K.H. conducted HAI assays with contribution of M.Z. to both areas. R.S.T., A.S.S.U., and M.C. conducted all other experiments. V.A.N., and T.B. conducted transcriptomic analyses with contributions from I.C.S., X.R-R., and E.S.C. R.S.T. conducted all other analyses with contribution from all authors. R.S.T. wrote the manuscript with contributions from all authors.

## Competing interests

The authors declare the following financial interests/personal relationships which may be considered as potential competing interests: V.A.N., T.B., I.C.S., X.R-R., and E.S.C. are employees of AstraZeneca and may hold stock or stock options. AstraZeneca are the manufacturers of Fluenz Tetra/FluMist Quadrivalent intranasal influenza live virus vaccine. K.M.P. is on the data safety monitoring board for two vaccine studies (NCT05249829, NCT05575492) and has received a fee for speaking from Seqirus and Sanofi Pasteur. P.J.M.O. has received fees for scientific advisory boards from GSK, Moderna, Seqirus, Janssen and Sanofi Pasteur. The remaining authors declare no competing interests.
