## [Peer Review File · Nature Communications]

Early mucosal events drive distinct mucosal and systemic antibody responses to live attenuated influenza vaccineEditorial Note: This manuscript has been previously reviewed at another journal that is not operating a transparent peer review scheme. This document only contains reviewer comments and rebuttal letters for versions considered *at Nature Communications*.

REVIEWER COMMENTS

Reviewer #1 (Remarks to the Author):

This reviewer's central criticism was that the assertion that adaptive immune reactions are known to be compartmentalized between mucosal vs the periphery and this is associated with differential antigen drainage to mucosal associated vs non-mucosal LNs.

The response "The manuscripts highlighted by the reviewer are excellent but it is often assumed by vaccinologists (who are not primarily mucosal immunologists) that antibody responses to mucosal vaccines should be evident in the blood." is not a scientifically compelling answer.

The authors then still fail to cite the literature to account for the fundamental forces at play that intrinsically separate mucosal vs non-mucosal associated adaptive immunity, which is at the heart of what they are studying.

The reviewer would reiterate that the study is very well executed and the data should be published. However, this cannot be done while disregarding the above information.

Reviewer #2 (Remarks to the Author):

The authors have incorporated reviewer suggestions and the manuscript is improved.

I have 3 very minor editorial suggestions below:

1. The included discussion in response to reviewer 1, on LAIV shedding should include references.
2. When referring to "The small number of Non-Shedders in our analysis" the n= number

could be given in the text to highlight size limitations- what scale study would be needed to identify a robust group of non-shedders?

3. The resolution of some supplementary figures appears low and fuzzy- eg Fig S7- check

Reviewer #3 (Remarks to the Author):

Summary – The manuscript by Thwaites and colleagues describes the mucosal and systemic antibody and cellular response to live-attenuated influenza vaccine in young adults. The primary findings are focused on differential induction of nasal IgA versus serum IgG by vaccination and sequencing & cytokine data to attempt to uncover correlates of an antibody response. Correlates of protection induced by LAIV are a major area of need in the field, although the study falls short of achieving that. There is utility in the data provided and the study design is additive of the field, despite some of the major findings being more confirmatory.

Major Comments –

1) Impact – The major finding as reflected in the title regarding compartmentalization of antibody induction by LAIV is confirmatory of the current understanding in the field. In addition, relationships between interferon induction and antibody response to LAIV have also been reported. In general there is very limited discussion of existing work in the field to examine immune induction by LAIV. Observations related to cytokines and immune cells are of interest and do advance the field.

2) Demographics – The authors should include a demographic table and information about previous vaccination for their cohort. The observation that LAIV induced serum Ab increases for all 4 strains is at odds with the majority of studies in the field that have shown very limited serum antibody response. Seroconversion to LAIV is also very high compared to most studies. The authors should discuss these differences and unique aspects of the cohort.

3) Children vs. Young Adults – The authors tend to interchangeably discuss data with LAIV in children and the data in this study. It is highly likely that immune response in pre or peri-

pubescent children differ from young adults. Data from the current study cannot be directly extrapolated to children.

Minor Comments –

1) Conclusion Line 254 – The authors conclude that early IL-33 response does not affect subsequent IFN responses to LAIV. There are no data to support this conclusion.

2) Conclusion Line 390 – This summation is over-reaching and contradicts the statement on Line 254 about IL-33 and interferon being independent.

REVIEWER COMMENTS

Reviewer #1 (Remarks to the Author):

This reviewer's central criticism was that the assertion that adaptive immune reactions are known to be compartmentalized between mucosal vs the periphery and this is associated with differential antigen drainage to mucosal associated vs non-mucosal LNs.

The response "The manuscripts highlighted by the reviewer are excellent but it is often assumed by vaccinologists (who are not primarily mucosal immunologists) that antibody responses to mucosal vaccines should be evident in the blood." is not a scientifically compelling answer.

The authors then still fail to cite the literature to account for the fundamental forces at play that intrinsically separate mucosal vs non-mucosal associated adaptive immunity, which is at the heart of what they are studying.

The reviewer would reiterate that the study is very well executed and the data should be published. However, this cannot be done while disregarding the above information.

We apologise to the reviewer for our incomplete revision of the text in response to this central point. On further revision we have separated a paragraph of the discussion to specifically introduce the immunological processes that mediate compartmentalization. This is a considerable topic, but we have highlighted the key principles and referenced key texts in the field within word count limitations, including those highlighted by the reviewer and a recent *Nature Reviews Immunology* piece (PMID: 34312520) so the interested reader can more completely understand the context of the work. New paragraph:

“Immune responses in the respiratory mucosa are compartmentalized from those of the systemic immune system and other mucosal tissues, with local responses dependent on mucosa-associated lymphoid tissues [13, 14]. The immunological processes underpinning compartmentalization of local and systemic responses are thought to depend on antigen drainage to mucosal or non-mucosal lymphoid tissues and the environmentally influenced phenotypic and functional distinctions of mucosal lymphocytes [13, 14, 15].”

We thank the reviewer for reiterating this important deficiency in the previous version, we believe that revising this area will provide the reader with a far better understanding of the context of our work.

Reviewer #2 (Remarks to the Author):

The authors have incorporated reviewer suggestions and the manuscript is improved.
I have 3 very minor editorial suggestions below:

1. The included discussion in response to reviewer 1, on LAIV shedding should include references.

We thank the reviewer for highlighting our lack of appropriate referencing in this section. This has now been resolved with the inclusion of the following references in this text section:

- Mohn, K.G., Smith, I., Sjursen, H. & Cox, R.J. Immune responses after live attenuated influenza vaccination. *Hum Vaccin Immunother* 14, 571-578 (2018). (reference 7)
- Shannon, I., White, C.L. & Nayak, J.L. Understanding Immunity in Children Vaccinated With Live Attenuated Influenza Vaccine. *J Pediatric Infect Dis Soc* 9, S10-S14 (2020). (reference 38)
- Jackson, D. *et al.* Viral Shedding in Recipients of Live Attenuated Influenza Vaccine in the 2016-2017 and 2017-2018 Influenza Seasons in the United Kingdom. *Clin Infect Dis* 70, 2505-2513 (2020). (reference 22)
- Costa-Martins, A.G. *et al.* Prior upregulation of interferon pathways in the nasopharynx impacts viral shedding following live attenuated influenza vaccine challenge in children. *Cell Rep Med* 2, 100465 (2021). (reference 46)

Text:

“Replication of LAIV, evidenced by viral shedding, has been considered an important determinant of vaccine efficacy [7, 38]. However, an increasing weight of literature supports the principle that detectable viral shedding may not be a pre-requisite for the induction of protective immunity [22, 46]...”

2. When referring to "The small number of Non-Shedders in our analysis" the n= number could be given in the text to highlight size limitations- what scale study would be needed to identify a robust group of non-shedders?

We thank the reviewer for highlighting this. We have add the n= number of Non-Shedders to the relevant section of text.

We appreciate the reviewers suggestion of adding a comment on the study size required to specifically study differences in mucosal responses between Shedder and Non-Shedders. We have now added the following text to the Discussion on this point:

“One fifth of participants in our study were Non-Shedders and further studies of immune responses in this group are warranted. Assuming consistency in the ratio of Shedders to Non-Shedders, a study size of 127 participants would be required to observe significant differences in nasal CXCL10 levels at 72 hours post-inoculation.”

3. The resolution of some supplementary figures appears low and fuzzy- eg Fig S7- check

We thank the reviewer for highlighting this. We have updated the supplementary figures to be higher resolution versions and will ensure these are maintained into the final publication pdf version.

Reviewer #3 (Remarks to the Author):

Summary – The manuscript by Thwaites and colleagues describes the mucosal and systemic antibody and cellular response to live-attenuated influenza vaccine in young adults. The primary findings are focused on differential induction of nasal IgA versus serum IgG by vaccination and sequencing & cytokine data to attempt to uncover correlates of an antibody response. Correlates of protection induced by LAIV are a major area of need in the field, although the study falls short of achieving that. There is utility in the data provided and the study design is additive of the field, despite some of the major findings being more confirmatory.

Major Comments –

1) Impact – The major finding as reflected in the title regarding compartmentalization of antibody induction by LAIV is confirmatory of the current understanding in the field. In addition, relationships between interferon induction and antibody response to LAIV have also been reported. In general there is very limited discussion of existing work in the field to examine immune induction by LAIV. Observations related to cytokines and immune cells are of interest and do advance the field.

As per our response to Reviewer 1, we have amended the manuscript to better highlight the existing understanding of compartmentalisation of immune responses at mucosal sites, including the following addition to the Introduction:

“Immune responses in the respiratory mucosa are compartmentalized from those of the systemic immune system and other mucosal tissues, with local responses dependent on mucosa-associated lymphoid tissues [13, 14]. The immunological processes underpinning compartmentalization of local and systemic responses are thought to depend on antigen drainage to mucosal or non-mucosal lymphoid tissues and the environmentally influenced phenotypic and functional distinctions of mucosal lymphocytes [13, 14, 15].”

In response to the reviewers second comment on Demographics (below), we have also added to the Introduction and Discussion to position our results alongside those of the literature on this field.

2) Demographics – The authors should include a demographic table and information about previous vaccination for their cohort. The observation that LAIV induced serum Ab increases for all 4 strains is at odds with the majority of studies in the field that have shown very limited serum antibody response. Seroconversion to LAIV is also very high compared to most studies. The authors should discuss these differences and unique aspects of the cohort.

We thank the reviewer for highlighting these shortcomings and the opportunity to improve our manuscript. We have added a demographics table to the Supplementary Materials file as suggested (new Table S1). The methods section for the manuscript details the exclusion criteria used for the study, which should also assist the reader in understanding the nature of the cohort, including exclusion based on receipt of an influenza vaccine in the last 2 years, clinically diagnosed influenza, or a long-term health problem that would trigger a recommendation of influenza immunisation.

We have expanded the discussion of other studies looking at seroconversion responses in similar adult groups, particularly Barria, M.I. et al. Localized mucosal response to intranasal live attenuated influenza vaccine in adults. *J Infect Dis* 207, 115-124 (2013), (reference 18), with the Introduction now stating:

“LAIV vaccination has been documented to result in blood antibody responses against H1N1 in just 9% of participants when measured by HAI or 24% of participants when measured by immunohistochemistry, while nasal IgA responses were seen in 33% of participants [18]”

The results of Barria et al are similar to our results, where 45% of participants had a >4-fold rise in serum IgG to at least one LAIV component, with 20% of the cohort having a >4-fold rise in serum IgG to H1N1. This closely aligns with the 24% response to this virus reported by Barria et al using different techniques. Similarly, for responses to HAI we observed 4/40 (10%) participants with a >4-fold rise in HAI to H1N1, very similar to the 9% reported by Barria et al. We have included these points in the Discussion through the following addition (in bold in the revised manuscript file):

“One analysis of humoral responses of adults to LAIV noted that 9% of participants had >4-fold serum HAI titre rises to H1N1 and 24% of participants had increased binding antibodies against H1N1, measured by immunohistochemistry [18]. These results closely align with responses against H1N1 we observed, where 4/40 (10%) of our cohort had serum HAI responses and 8/40 (20%) had >4-fold rises in serum IgG to H1N1.”

We believe that this direct comparison to a larger published study in a similar demographic group supports our observed seroconversion rates, reassuring us that our results are typical of this age group. We thank the reviewer for highlighting this opportunity to improve the manuscript.

3) Children vs. Young Adults – The authors tend to interchangeably discuss data with LAIV in children and the data in this study. It is highly likely that immune response in pre or peri-pubescent children differ from young adults. Data from the current study cannot be directly extrapolated to children.

We agree that the results of the present study cannot be assumed to translate directly to responses in children. We have revised the text throughout to specifically mention the age group of study in each reference. We have also added an explicit sentence to the Discussion to highlight this issue:

“Furthermore, our results in young adults cannot be assumed to apply to children or older adults.”

Minor Comments –

1) Conclusion Line 254 – The authors conclude that early IL-33 response does not affect subsequent IFN responses to LAIV. There are no data to support this conclusion.

Our correlation matrix analysis of key nasal mediator results (Figure 2G) demonstrates that the scale of the IL-33 response at 8h is not correlated with the peak (72h) IFN response. However, on revision we noted that the previous text in this section inferred a direct causal relationship between these observations, which the reviewer correctly highlights as an unjustified assumption. We have therefore amended the text in this section to highlight as association, rather than a causal relationship (amended text in bold):

“Thus, an IFN-related response to LAIV is evident in the nasal mucosa following LAIV inoculation, preceded by an earlier release of IL-33 which is not associated with the subsequent IFN-response but is inversely related to viral replication.”

2) Conclusion Line 390 – This summation is over-reaching and contradicts the statement on Line 254 about IL-33 and interferon being independent.

We thank the reviewer for highlighting this discrepancy with the results in Figure 3D (and discussed in Minor Comment 1). We have amended the text in line 390 (now line 400) as follows (amended text in bold):

“While IL-33 induction at 8h p.i. was equivalent between antibody response groups (data not shown), the inverse association between IL-33 release and the subsequent scale of vaccine shedding (Fig. 2H-I) may indicate that early alarmin release suppresses viral replication **via mechanisms independent on the subsequent** IFN-led mucosal anti-viral immune responses that in turn associate with distinct mucosal and peripheral antibody responses to LAIV in healthy young adults (Fig. 7).”

REVIEWERS' COMMENTS

Reviewer #1 (Remarks to the Author):

outstanding issues now addressed

Reviewer #3 (Remarks to the Author):

The authors have addressed my concerns.